# Bottom-up Instance Segmentation of Catheters for Chest X-ray Images

**Francesca Boccardi**[1,2]                    FRANCESCA.BOCCARDI@STUDIO.UNIBO.IT

**Axel Saalbach**[2]                    AXEL.SAALBACH@PHILIPS.COM

**Heinrich Schulz**[2]                    HEINRICH.SCHULZ@PHILIPS.COM

**Samuele Salti**[1]                    SAMUELE.SALTI@UNIBO.IT

**Ilyas Sirazitdinov**[2]                    ILYAS.SIRAZITDINOV@PHILIPS.COM

[1] *University of Bologna, Bologna, Italy*

[2] *Philips Innovative Technologies, Hamburg, Germany*

**Editors:** Accepted for publication at MIDL 2024

## Abstract

Chest X-ray (CXR) is frequently used in emergency departments and intensive care units to verify the proper placement of central lines and tubes and to rule out related complications. The automation of the X-ray reading process can be a valuable support tool for non-specialist technicians and minimize reporting delays due to non-availability of experts. While existing solutions for automated catheter segmentation and malposition detection show promising results, the disentanglement of individual catheters remains an open challenge, especially in complex cases where multiple devices appear superimposed in the X-ray projection. In this paper, we propose a deep learning approach based on associative embeddings for catheter instance segmentation, able to effectively handle device intersections.

**Keywords:** chest X-ray, instance segmentation, catheters, tubes, CVC, SWG

## 1. Introduction

Chest X-ray (CXR) is widely used to detect lung abnormalities and pathologies. In emergency departments and intensive care units, CXRs are often employed to check the correct placement of medical devices in order to exclude potentially dangerous complications (Gambato et al., 2023). In those cases, a careful assessment of the CXR is required and is typically performed by the attending radiologist. The development of an automated process could assist non-professional technicians in the interpretation of X-rays when an expert is not available, potentially reducing reporting delays. Several efforts have been made to deploy automated solutions for catheter segmentation (Subramanian et al., 2019), malposition recognition (Hansen et al., 2021), and catheter tip detection (Jung et al., 2022). In addition, it is not rare that a single CXR may contain multiple devices, often appearing as superimposed in the projection image. In such scenarios, it is essential to identify and separate individual catheters, which is still an open problem. The use of conventional methods, such as Mask R-CNN (He et al., 2017), which rely on a detect and segment approach, may be limited when dealing with thin and long catheters, that often extend through the entire image. In this study, we propose a deep learning approach based on associative embedding for cardiac catheter instance segmentation. The key contributions of this study can be summarized as follows: adaptation of the LaneNet (Neven et al., 2018) architecture for catheter instance segmentation, by using HRNet(V2) (Sun et al., 2019) as a customized backbone; an intersection resolution algorithm to effectively handle catheters intersections.

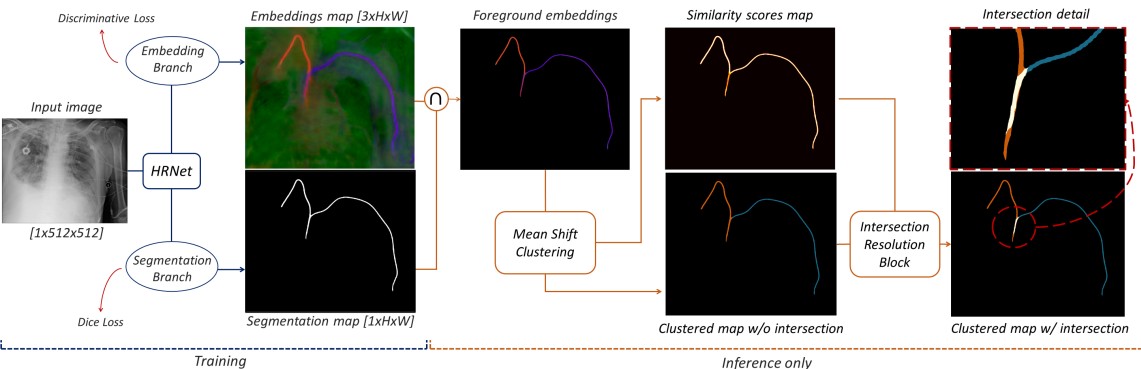

Figure 1: Schematic representation of the proposed method. An adapted HRNet(V2) architecture is used to predict segmentation masks and embeddings. To disentagle catheters instances, mean shift clustering is performed on segmented pixels embeddings and the intersection areas are then detected.

## 2. Methods

Our method was inspired by the LaneNet architecture (Neven et al., 2018), which was originally designed to solve a road lane detection task, by treating it as an instance segmentation problem. Similarly, we aim to segment cardiac catheters which are long and thin structures. We employ a branched multi-task network, consisting of two parts: a segmentation branch that detects the pixels belonging to a catheter, and an embedding branch that predicts an embedding value for each of them. To disentangle segmented pixels into different catheters instances, we perform clustering on their embeddings. After that, we employ an algorithm to detect intersection pixels and assign them to all the involved catheters (Figure 1).

**Segmentation and embeddings prediction.** In our implementation, we use an HRNet(V2) architecture as backbone, with two nearly identical branches: a segmentation branch, trained with the Dice loss to output a segmentation map, and an embedding branch, trained with the discriminative loss (Neven et al., 2018) to map each segmented pixel to a point in a 3d feature space, called pixel embedding. In this embeddings space, pixels belonging to the same catheter end up close together, while pixels belonging to different instances lie far apart. The discriminative loss achieves this by pulling embeddings of the same cluster towards the mean embedding, while pushing different clusters further apart.

**Embeddings clustering.** During the inference, the network generates a semantic segmentation map, identifying which pixels are part of a catheter, and a 3d embedding for each segmented pixel. We then use the segmented pixels embeddings in a mean shift clustering algorithm to assign each pixel to an individual catheter.

**Intersection resolution.** In our images, different catheters may be superimposed in the projection image. To properly detect intersections, we developed an intersection resolution algorithm. While mean shift simply splits the set of foreground pixels into $n$ groups, we observed that, in the embeddings space, the intersection pixels often lie between the corresponding clusters centers (catheters). Therefore, starting from a mean shift partition, for each pixel we compute a similarity score $s$, that measures how similar, i.e. close, the embedding is to its assigned cluster center compared to the other cluster centers determined by mean shift. If $s$ is lower than some threshold, the pixel is considered to belong to the intersection of the clusters (catheters) involved, so that it is assigned to both.

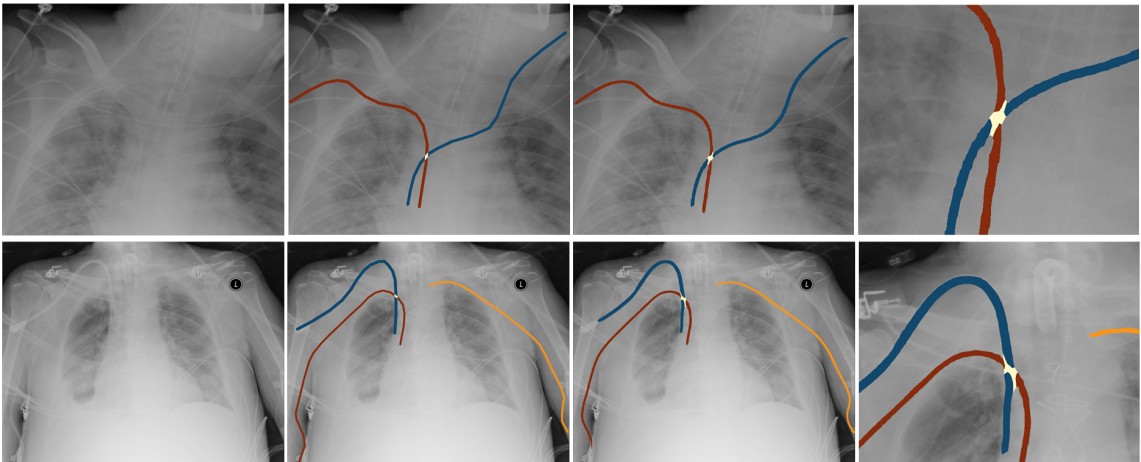

Figure 2: From left to right: CXR, ground truth, our prediction, detailed prediction and intersection.

| Model | IoU | Dice | AP | AR |
|---|---|---|---|---|
| HRNet(V2) + CC | $0.608 \pm 0.004$ | $0.746 \pm 0.004$ | $0.149 \pm 0.013$ | $0.174 \pm 0.015$ |
| Ours | $0.599 \pm 0.010$ | $0.739 \pm 0.009$ | $\mathbf{0.726 \pm 0.013}$ | $\mathbf{0.807 \pm 0.013}$ |

Table 1: Semantic (IoU, Dice) and instance (AP, AR) segmentation results of our method and the baseline.

## 3. Experiments and results

We used 8877 CXRs from the RANZCR CLiP dataset (Tang et al., 2021), with 11786 annotated Central Venous Catheters (CVC) or Swan-Ganz Catheters (SWG). Every CXR has at least 1 device, with 30% of the images containing between 2 and 4 devices. We trained a branched HRNet(V2-W30) architecture pre-trained on ImageNet, resizing images to 512x512px and performing a patient-stratified 5-fold cross-validation. Hyperparameters in the training and inference recipes were optimized on the validation set. Figure 2 shows successfully processed CXRs with the resolved intersections. To overcome the lack of a baseline from the literature, we compare our method to a semantic segmentation HRNet(V2), followed by morphological processing via connected component analysis (CC) on predicted binary segmentation maps to identify isolated regions as instances of catheters. We evaluate the results using IoU and Dice for semantic segmentation, and average precision (AP) and average recall (AR) for instance segmentation (Table 1). AP and AR are averaged over IoU thresholds in a range of [0.2,0.6], chosen based on a reader study [1] that evaluated the quality of manual CVC annotations. Our method provides significantly better AP and AR values than the baseline approach, which is prone to over- and under-segmentation issues.

## 4. Conclusion

Our bottom-up approach based on associative embeddings was able to effectively perform instance segmentation on chest radiographs with multiple cardiac catheters. To the best of our knowledge, it is the first successful application of deep learning for catheter instance segmentation.

---

1. AP340 KI-RAD, KI-SIGS Summit 2022, Lübeck, Germany

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
