# OpenReview forum: "Bottom-up Instance Segmentation of Catheters for Chest X-ray Images"
_MIDL.io/2024/Short_Papers — MIDL 2024 Short Papers_

### Official Review · Reviewer_guuN · 2024-04-23

**Confidence:** 5
**Final Rating:** 5

**Review:**

This paper tackles instance segmentation of catheters in chest X-rays. The paper is well-written, results are excellent, and the task and approach are interesting for the MIDL audience.

Strengths
- The paper sets a new benchmark for a challenging task.
- Authors use publicly available data.
- Authors show that they achieve comparable results on semantic segmentation as a baseline method, and significantly outperform the baseline method in terms of instance segmentation.

Weaknesses
- The work appears to be heavily based on instance segmentation in road lane detection, limiting the methodological contribution of this approach.

---

### Decision · Program_Chairs · 2024-04-26

Accept